# Memo: Training Memory-Efficient Embodied Agents with Reinforcement Learning

**Gunshi Gupta**[*]
University of Oxford

**Karmesh Yadav**
Georgia Tech University

**Zsolt Kira**
Georgia Tech University

**Yarin Gal**
University of Oxford

**Rahaf Aljundi**
Toyota Motor Europe

## Abstract

To enable embodied agents to operate effectively over extended timeframes, it is crucial to develop models that form and access memories to stay contextualized in their environment. In the current paradigm of training transformer-based policies for embodied sequential decision-making tasks, visual inputs often overwhelm the context limits of transformers, while humans can maintain and utilize a lifetime of experience compressed as memories. Significant compression is possible in principle, as much of the input is irrelevant and can be abstracted. However, existing approaches predominantly focus on either recurrent models with fixed-size memory or transformers with full-context reliance. In this work, we propose Memo, a transformer-based architecture and training recipe for reinforcement learning (RL) on memory-intensive, long-horizon tasks. Memo incorporates the creation and retrieval of memory by interleaving periodic summarization tokens with the inputs of a model during training. We demonstrate Memo's effectiveness on a grid-world meta-RL benchmark and a multi-object navigation task in photo-realistic indoor settings. Memo outperforms naive long-context transformer baselines while being more compute and storage efficient. Additionally, Memo generalizes better to longer contexts at inference time and remains robust in streaming settings, where historical context must be truncated to fit inference constraints.

## 1  Introduction

Humans intuitively prioritize and retain memories relevant to their current tasks, filtering out irrelevant details such as the color of a house's walls along a route unless it serves as a crucial navigation landmark. Similarly, reinforcement learning (RL) agents must learn to selectively retain and access task-specific memories guided by task objectives rather than predefined rules. This ability is especially critical for long-horizon tasks, where delayed rewards make credit assignment challenging and necessitate efficient training and inference over extended timescales to effectively capture the relationship between past and future events.

Recently, Transformers have become the go-to framework for sequence modeling due to their flexible and expressive encoding, free from the fixed-size constraints of recurrent neural networks (RNNs) [8, 14]. However, it assumes access to all past information at each step rather than selectively retaining relevant memories, which can lead to inefficiencies and make it harder to extract task-relevant information for decision-making. Their quadratic attention complexity further limits scalability, making it challenging to process long contexts efficiently, especially when gradients must propagate over large sequences. At test time, Transformers face another practical challenge: storing an

---

[*]Correspondence to: `gunshi.gupta@lmh.ox.ac.uk`

39th Conference on Neural Information Processing Systems (NeurIPS 2025).

increasingly large key-value (KV) cache is computationally expensive, and attending over ever-growing contexts requires the ability to generalize beyond the temporal patterns seen during training.

To address these challenges, we propose **Memo**, a novel framework that enhances transformer-based agents by enabling them to summarize and index past experiences through specialized summary tokens. Memo introduces a simple yet effective training procedure where transformers learn to compress their experience by periodically generating summary tokens that encode relevant past information. These summary tokens are stored in a dedicated memory buffer, allowing the model to attend to condensed representations of prior states instead of maintaining a full-context cache. This enables Memo to maintain a compact memory footprint at inference, reducing computational costs while preserving long-horizon reasoning capabilities.

Inspired by advancements in extending context lengths in language models, Memo adapts these principles to reinforcement learning (RL), addressing the unique challenges presented by RL tasks. Specifically, Memo integrates with both on-policy and off-policy RL agents, leveraging learned compression mechanisms to enhance sample efficiency and generalization in sequential decision-making settings. We demonstrate its effectiveness across diverse tasks, including a grid world and a 3D indoor navigation task in Habitat. Memo matches or outperforms the naive transformer baseline, which requires storing its entire context, leading to an 8-10× larger cache. Despite using significantly less memory, Memo achieves better in-context learning (ICL) behavior, higher success rates, and shorter path lengths in unseen environments. On large-scale navigation tasks, we show that Memo also exhibits superior robustness in streaming settings where the cached context is truncated, demonstrating improved adaptability under limited inference budgets.

Our contributions are summarized as follows:

- We propose Memo, a framework that enables transformers to learn to summarize and store task-relevant past experiences, reducing computational costs while maintaining long-horizon reasoning capabilities.
- We evaluate Memo on sequential decision-making tasks, demonstrating its efficiency and improved in-context learning behavior compared to transformers that require full-context storage.
- We show that Memo is a general and versatile memory augmentation technique applicable to both on-policy and off-policy RL agents for long-horizon tasks.

## 2 Related Work

In this section we briefly review research on extending context limits of transformers in language modeling, as well as the state of the art in long sequence modeling and decision making in RL.

**Extending Transformer Contexts in language**: A significant limitation of current large language models is their restricted input context length, prompting research into methods for scaling and generalizing to larger contexts [33, 21, 13, 27]. Key-value (KV) caching can reduce recomputation at training [9] or inference time [23], by storing and reusing self-attention outputs during decoding. [25] further reduce memory by maintaining a compressed cache, though this compression is not completely task-guided.

Beyond compute efficiency, recent work explores dynamic context extension. Recurrent Memory Transformer (RMT) [4] introduces summarization tokens to periodically compress and propagate prior context while discarding older tokens. Autocompressors (AC) [7] extend this by accumulating generated summaries across context windows, avoiding RMT's fixed-size memory but still truncating gradient propagation through summaries. We extend this context summarization approach to the more involved setting of reinforcement learning (RL), where summarization must support decision-making and handle credit assignment over long horizons. Unlike AC, which fine-tunes pretrained models on a supervised token prediction task with limited gradient propagation—and aims to match but does not surpass full-context transformer baselines—we train from scratch, integrating summarization directly into the RL optimization process and propagating gradients across all summaries. This enables more effective task-driven memory formation, surpassing full-context transformer baselines in both efficiency and scalability.

**Transformers in RL**: Transformers, while widely used in self-supervised language modeling [10, 3], have seen slower adoption in reinforcement learning (RL) due to challenges like sparse rewards, optimization instability, and limited batch sizes in on-policy RL. However, as we move to more

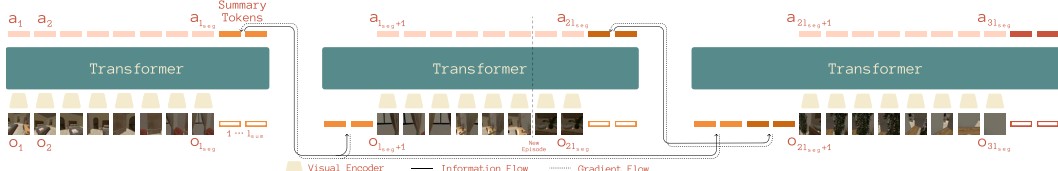

Figure 1: **Architecture Diagram of Memo.** The frames $O_{1-3l_{seg}}$ depict the input sensor observations to the agent at timesteps $1 - 3l_{seg}$. The figure depicts the information flow between three consecutive segments of a much longer context of inputs provided to a transformer during training. The summary tokens bottleneck the information passed on from one chunk of inputs to the next.

complex tasks that demand long-horizon planning and reasoning over past experiences, transformers have shown promising initial results, outperforming recurrent models in POMDPs when encoding longer temporal dependencies [22].

To stabilize transformer training in partially observable RL, Parisotto et al. [24] used KV caching while restricting gradient propagation to a small window of time steps, limiting the learning of long-term dependencies. More recently, ReLIC [11] improved convergence of on-policy RL algorithms by unfreezing the historical context within a rollout and applying frequent updates, while AMAGO [12] adapted off-policy RL to train with a shared transformer backbone. While these methods enhance long-horizon learning, ReLIC's high computational cost and slow training hinder scalability, and AMAGO is limited to small state-space observations, reducing its applicability to more complex tasks. Our approach addresses these challenges by enabling efficient memory utilization and leveraging a more scalable transformer architecture for extreme long-horizon decision-making.

## 3 Methodology

This section outlines our methodology for memory-augmented in-context reinforcement learning (RL). We first formalize the problem and describe the in-context RL algorithms and the sequence model we consider, in Section 3.1. We then introduce our context summarization mechanism in Section 3.2, and describe key implementation details including the attention masking, positional encoding, segment randomization and cache management scheme we use.

### 3.1 Problem Setting

In this work, we study tasks that require a model to perform sequential decision-making over long executions while leveraging its experience history to improve efficiency over time. Many diverse reasoning problems fall into this category, ranging from embodied navigation to language modeling. We formalize these tasks within the framework of in-context reinforcement learning (RL), where models adapt and improve their performance using past inferences rather than pre-collected supervised datasets. By treating these tasks as partially observable Markov decision processes (POMDPs) and transforming them into fully observable MDPs through the integration of historical context, it is possible to train policies effectively using both on-policy and off-policy RL methods.

We use a sequence encoder to model the agent's policy, processing past observations into a contextual representation for decision-making. At each timestep $t$, the sequence encoder receives a sequence of observations, denoted as $X_t = \{o_1, \ldots, o_t\}$, and maps them to hidden representations $h_t = \text{SeqEnc}(X_t)$. The policy distribution and value estimate are then computed using learnable actor and critic heads as $\pi(a_t \mid h_t), V(h_t)$, respectively, where $a_t \in \mathcal{A}$ is the action taken by the agent at timestep $t$, selected according to the policy $\pi$ over the action space $\mathcal{A}$.

**In-Context RL Algorithms**: In in-context RL, an agent adapts over time by conditioning on its entire past within a trial. A trial consists of multiple episodes, where each episode resets upon termination, but the agent retains memory across episodes. These episodes share a common underlying context—such as environment dynamics, task structure, or goal distribution—enabling the model to leverage past experiences for more efficient adaptation. Sequence models like Transformers encode history by attending to all previous timesteps within a trial, allowing the agent to infer temporal dependencies and adjust its behavior accordingly. However, as trial lengths increase, attending to all past timesteps becomes computationally infeasible.

To address this, we introduce Memo, a method for constructing and accessing memory representations in Transformers to improve long-term context management in RL, as described in Section 3.2. To demonstrate its efficacy and broad applicability to long-context RL, we integrate Memo with both on-policy and off-policy RL method:

- On-policy RL: We incorporate our memory mechanism into RELIC [11], an adaptation of DD-PPO [30], which applies frequent partial updates during trajectory rollouts to help transformer policies learn more effectively from in-context experiences.
- Off-policy RL: We extend Memo to AMAGO [12], which trains transformer-based policy using a shared sequence encoder and unified actor-critic loss to improve stability and learning efficiency.

### 3.2 Context Summarization in Memo

Following [7], we introduce context summarization as a learned sub-task that enables the transformer encoder to compress and retain task-relevant information from long observation histories. This mechanism allows the model to efficiently process past experiences while optimizing for task rewards in RL. Our approach uses learnable summary embeddings to prompt the transformer to generate summary tokens at predefined intervals, ensuring efficient memory utilization in long-horizon tasks.

Instead of attending to the full sequence history, the transformer periodically compresses past context into summary tokens, which are then fed back into the model in future timesteps. We partition long input sequences into segments of length $l_{seg}$, and generate $l_{sum}$ summary tokens at the end of each segment. These tokens act as compact memory representations, allowing the model to condition on past experiences without requiring access to raw observations. The encoding and integration of summary tokens into future segments are illustrated in Figure 1, and the pseudocoe is provided in Appendix A.4. The summarization mechanism is trained end-to-end through the RL objective, allowing the model to attend over and refine all previously generated summary tokens. Gradients propagate through the attention mechanism, ensuring that memory updates remain task-driven.

**Attention Masking**: We use causal masking which includes all the previous summary tokens as well as the previous observations within the segment being processed at time step $t$. This excludes any previous observation that contributed to and that was processed before the most recent summarization. This creates a bottleneck for the flow of historical information to be solely through summary tokens.

**Positional Encoding**: To add positional awareness, we assign indices to an observation at time step $t$ based on its relative position within a segment. We use the same positional encoding method (linear or ROPE) used by the baseline implementation which we will be comparing to. The position labeling scheme in a naive full-context transformer input has indices which run from 0 to the length of the context window. In Memo, the input context at time step $t$ in the rollout consists of $n * l_{sum}$ summary tokens and the latest segment's observations, where $n = \lfloor t/l_{seg} \rfloor$. The position indices thus range from 0 to $n * l_{sum}$ - 1 for the summaries and then start from $n * l_{sum}$ and go to $t - n * l_{seg} + n * l_{sum}$ for the most recent observations in the latest segment. The summary embeddings at the end of a segment are assigned indices following those of the last segment observation.

**Segment length randomization**: Following Chevalier et al. [7], we randomize segment lengths during training by sampling uniformly within $\pm 20\%$ of a fixed $l_{seg}$. This approach, which previously showed small improvements in perplexity for token prediction, will be ablated in 4.8. A fixed $l_{seg}$ is used during data collection and evaluation, while training uses randomized segment lengths.

**Maintaining the KV Cache**: In on-policy RL, the KV cache stored during action selection can become stale after a model update, as the new weights would encode different representations for the same past context. This divergence is problematic since policy decisions depend on context, making consistency crucial. ReLIC addresses this by refreshing and re-encoding the KV cache with the latest model weights after every on-policy update. We extend this approach to memory tokens, ensuring consistency by recomputing the summary vectors alongside the KV cache refresh. This guarantees that both cached representations and learned memory remain aligned with the current policy.

## 4 Experiments

In this section, we present experimental results highlighting different aspects of our proposed method. We first describe the benchmark tasks and baselines used in our experiments, followed by different experimental insights in the following subsections.

## 4.1 Benchmarks

**EXTOBJNAV**: The Extended Object Navigation (EXTOBJNAV) task, first introduced in [11], builds on the OBJECTNAV task commonly used in embodied AI research [1, 20]. While OBJECTNAV requires an agent to navigate to a single object goal in an episode, EXTOBJNAV focuses on how performance evolves when the agent is repeatedly tasked with reaching different object goals within the same scene. The task evaluates the agent's ability to leverage memory—utilizing information from prior exploration during navigation to reach subsequent goals.

The EXTOBJNAV task uses 37 training and 12 validation scenes from HSSD [16] and includes 20 object instances from the YCB dataset [6]. These objects can be randomly placed on receptacles throughout the scene, with each *placement* featuring an average of 30 objects. We use 11,100 novel placements during training and 108 during evaluation. A sample scene is shown in Figure 2a.

Agents are trained and evaluated over *trials*, each consisting of a sequence of episodes with the same object placement in a scene. Each episode mirrors OBJECTNAV in structure: the agent is randomly spawned and tasked with locating an object from a sampled category. Episodes are capped at 500 steps, while trials are limited to 4096 steps during training and 32k steps during evaluation.

The agent is a Fetch robot with a head-mounted camera (256x256 RGB) and an odometry sensor that provides relative position and orientation. It operates in a discrete action space: Forward (0.25m), TurnLeft (0.3°), TurnRight, and Stop. An episode is successful if the agent stops within 2m of the goal with the object in view, occupying at least 10 pixels in the image.

We report success rate (SR) and success weighted by path length (SPL), with SPL measuring navigation efficiency relative to the shortest path. Results are computed over 32k interaction steps on the validation set and plotted at intervals of 500 steps. All experiments use 10 random seeds. Further details are in Appendix A.1. We include results on another multi-goal 3D maze navigation task (Memory Maze) in the appendix.

**Dark-Key-To-Door**: Dark-Key-To-Door [18] is a Meta-RL [2] benchmark where an agent must find an invisible key to open an invisible door in a 9x9 2D gridworld. The agent receives +1 reward for finding the key and the door. It only observes its (x, y) position at each timestep and must remember key and door locations based on previous reward signals.

Each episode lasts up to 50 timesteps, with trial duration fixed at 500 steps. We evaluate performance by average reward across 960 trials and 3 validation seeds. An agent that leverages contextual information can complete multiple episodes per trial and achieve total reward well above 50.

## 4.2 Baselines

Below, we describe the main transformer-based baselines that we compare Memo to in this work.

- The **full context transformer** baseline corresponding to ReLIC [11] and AMAGO [12]. We refer to it as FCT in the following sections for brevity.
- **Transformer without Inter-Episode Attention (no-IEA):** We modify the attention masking such that the model can't attend over previous trials in the rollout. This sets a lower bound for what the performance would be if the model didn't do in-context learning and simply used its current trial's observations to explore and find objects.
- **Recurrent Memory Transformers (RMT):** We also include a recurrent-only version of Memo that does not accumulate summary tokens. This baseline represents an implementation of RMT [5] adapted for the RL setting. The purpose of this comparison is to evaluate the benefits of Memo's summary accumulation against a fixed-size memory constraint. To ensure a fair comparison, we set the fixed memory size in RMT equal to the total number of summary tokens Memo would accumulate over multiple segments ($l_{sum}$ * number-of-segments).
- **Autocompressors (AC):** Memo with a pre-trained initialization and highly truncated backpropagation through time TBTT. This represents an implementation of AC [7] tailored to the RL setting.

We exclude $RL^2$ and Tr-XL due to their poor performance in the results of [11, 12]. In the following subsections, we present experimental insights examining how Memo's memory creation mechanism and training recipe enhance long-horizon reasoning and in-context learning. Each subsection highlights and focuses on a specific finding and analyzes a subset of baselines drawn from figures that are shared across multiple sections, allowing us to do a more structured comparison.

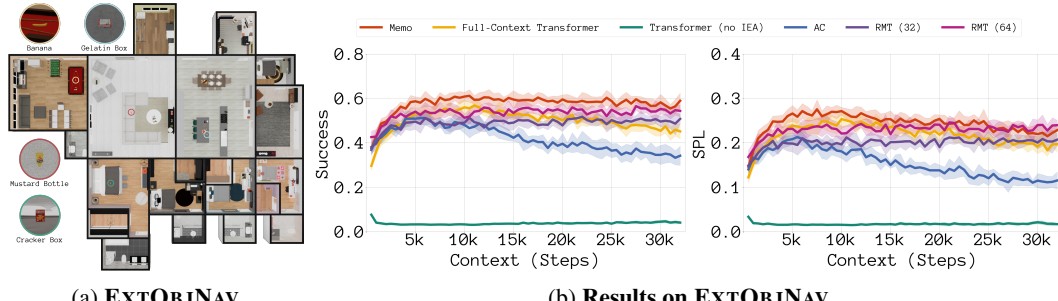

|  (a) EXTOBJNAV | (b) **Results on EXTOBJNAV** |

Figure 2: **(Left)** Overhead view of a training scene in Habitat simulator. The EXTOBJNAV task involves placing multiple objects around the house, and then sampling an object category as the goal each time the previous goal is reached. The figure shows some sample objects that may be placed around the house, such as Banana, Cracker Box, etc. **(Right)** Val success rate and SPL curves over 32k in-context learning steps in novel scenes, for different methods trained with ReLIC. We compare Memo to the the full-context transformer (FCT), the FCT variant which does not attend over previous episodes (no IEA), the recurrent memory transformer (RMT), and the Autocompressors (AC) variant.

## 4.3 Summarization Outperforms Full In-Context Access

We first show that Memo achieves higher performance and exhibits better ICL-ability compared to naive full context transformer (FCT) while using significantly less tokens in-context during evaluation.

In Figure 2b, we report the validation SR and SPL on the EXTOBJNAV task over 32k environment steps. All methods are trained using on-policy RL technique proposed by ReLIC. Memo is trained with a segment length of 256, 32 summarization tokens, and a trainable rollout length of 4096. We observe that Memo outperforms FCT achieving 7.5% higher SR and 2.5% higher SPL on average while using 8x fewer tokens (details in Table 1). We see that both Memo and FCT show in-context learning ability and continue to increase their performance up to 10k steps (∼2.5x of the training context length). Beyond this point, both methods experience performance degradation due to limitations in context length extrapolation. Notably, FCT degrades more than Memo in SR, while both show similar degradation in SPL. The single-episode variant (Transformer - no IEA) performs significantly worse, highlighting the disadvantage of not leveraging historical context in this navigation task. We also compare the GPU memory and FLOPs for Memo and FCT in Appendix A.5, and their sample-efficiency in Appendix A.10.

In Figure 3a, we present a similar comparison on the Dark-Key-To-Door environment, reporting the average total return over trials as training progresses. All methods are trained using the off-policy RL algorithm AMAGO. We see that both Memo and RMT match the FCT baseline, achieving peak performance on validation episodes by 40M training steps. Interestingly, FCT shows a notable performance drop around the 35-40M step mark across all seeds, while Memo remains maintains stable convergence. This is likely due to training instability commonly observed in long-context RL [12, 24], highlighting further training challenges with full-context models.

The consistent performance gains observed across both on-policy (with ReLIC) and off-policy (with AMAGO) RL setups demonstrates that our proposal to augment transformer policies with summarization enhances long-context modeling capabilities while being independent of the specific RL algorithm used.

## 4.4 Advantage of Summary Accumulation Over Fixed Recurrent Memory

The summary accumulation mechanism in Memo addresses the limitations of both transformers and recurrent models. While transformers can model full context, they suffer from significant memory overhead with long sequences. In contrast, recurrent models are memory-efficient but struggle to propagate gradients effectively over long horizons due to vanishing or exploding gradients. Summary accumulation mitigates context explosion by periodically summarizing experiences and propagating these summaries forward, allowing earlier memories to directly influence outcomes at later timesteps. Additionally, in purely recurrent models, gradients must pass through all intermediate memory update steps to reach the embedding of a relevant input far back in time. In contrast, periodic summary

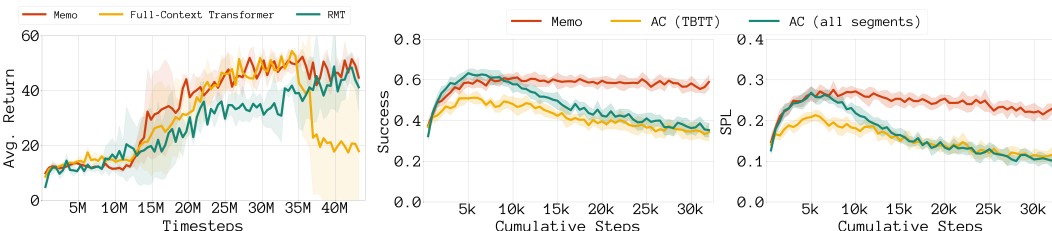

(a) **Results on Dark-Key-To-Door**  (b) **Comparison between Memo and AC variants on EXTOBJNAV.**

Figure 3: **(Left)** We plot the average return over evaluation trials of 500 steps, consisting of 10 or more episodes, plotted against the training progress. We compare Memo, RMT and FCT, when training with the AMAGO RL algorithm over 3 random seeds. **(Right)** On EXTOBJNAV, we find that restricting gradient propagation to a few consecutive windows (AC (TBTT)) performs much worse than allowing all past segments to remain trainable (AC (all segments)). While AC (all segments) initially performs slightly better than Memo, it suffers from severe performance degradation over time, converging to AC (TBTT) after 16k steps.

generation and accumulation (as in Memo) ensure that each summary vector contributes directly to the loss function at later timesteps without degradation. For example, at a timestep $t$, all summaries created before are available as an undiminished input to the attention and subsequent optimization process.

We empirically validate this by comparing Memo's summary accumulation to a variant where only newly generated summaries (at the end of the current segment) are propagated to the next segment. This variant is closely aligned to the Recurrent Memory Transformer (RMT) [5]. To ensure fairness, we allow the recurrent model a larger summary size. Even with this adjustment, Memo achieves faster convergence, improving training speed by over 10M steps on the Dark-Key-To-Door task (Figure 3a). We further evaluate RMT on EXTOBJNAV (Figure 2b) using $1\times$ and $2\times$ Memo's summary size (32, 64). Although RMT outperforms a purely recurrent baseline such as $RL^2$—showing that per-step memory updates are too limiting—it still lags Memo by about 5% success rate. Larger summaries (RMT-128) destabilize training. This indicates that periodic summarization (like in Memo or Memo-RMT) improves expressivity and stability over purely recurrent per-step memory updates, while Memo's accumulation mechanism further enhances both by introducing residual-like gradient shortcuts that enable efficient and stable long-horizon optimization (see Appendix A.7 for further analysis).

### 4.5 Importance of Long-Horizon Gradient Propagation: Comparison to Autocompressors

Since Memo's summarization approach is inspired by the Autocompressors (AC) method from the language modeling domain, we investigate how following the exact training setup from AC would perform in our setting. This involves pre-training a transformer policy on a shorter context length before fine-tuning it with summarization to extend its effective context length. We first train a baseline transformer with a context size of 1024 for 350M steps (half of the total training horizon for other baselines). This model serves as the weight initialization for our AC model, which is then fine-tuned for 350M steps with segment length $l_{seg} = 256$, summary length $l_{sum} = 32$, and a trainable rollout length of 4096. Following AC, we employ Truncated Backpropagation Through Time (TBTT) during training, restricting gradient propagation to summary tokens from only the two preceding segments; we refer to this variant as **AC (TBTT)**. For a fairer comparison to other methods, we also train a version of AC where gradients are propagated across all segments within the rollout, referred to as **AC (all segments)**. This setup is identical to Memo, except it's bootstrapped from a pre-trained model with a shorter context length.

We discover that this limited propagation is insufficient for tasks requiring long-horizon memory, as AC (TBTT) fail to capture dependencies spanning extended horizons. As illustrated in Figure 3b, AC (TBTT) exhibits poorer in-context improvement in object-finding tasks compared to AC (all segments). While AC (all segments) slightly surpasses Memo in SR and matches it in SPL during the first 6-8k steps, its performance degrades significantly over time, eventually converging to AC (TBTT) after 16k steps. This suggests that while a short context length pre-trained checkpoint can offer some initial advantage, it ultimately hinders effective generalization to longer contexts.

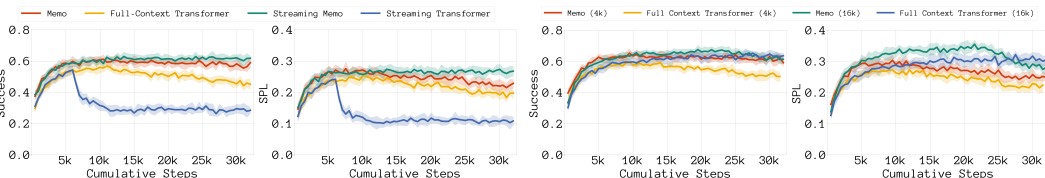

(a) **Memo and Transformer streaming evaluations.**      (b) **Finetuning for 16k steps on EXTOBJNAV.**

Figure 4: **(Left)** We show a comparison between the accumulating context (default setting) and the streaming version of Memo and the full-context transformer, where streaming starts after 6k eval steps. This comparison highlights the robustness of Streaming Memo, which maintains and even slightly improves performance, whereas Streaming Transformer suffers a sharp decline. **(Right)** We see that finetuning over a longer number of steps during training serves to partially fix the degradation due to context length extrapolation.

## 4.6 Extrapolation and Experience Subsampling During Evaluation

Avoiding memory explosion due to KV cache accumulation is a challenge for transformers at inference time. We evaluate Memo and the FCT baseline in a streaming setting, where only the most recent $T$ KV cache elements are retained for inference, enforcing a fixed memory limit. To keep our setup simple, we do not update the KV cache to account for shifts in token position indices relative to the retained cache. We set $T$ based on the context length at 6k env steps, as both Memo and FCT exhibit ICL generalization up to at least 6k steps (see Figure 2b). Thus, on EXTOBJNAV, we use $T = 6k$ for Streaming Transformer and $T = 1024(= round(6000/256) * 32 + 256)$ for Streaming Memo, since 6k steps correspond to 24 sets of summary tokens when $l_{sum} = 32$ and $l_{seg} = 256$.

We present the results in Figure 4a. Streaming Memo not only maintains performance through the trial but even outperforms Memo with full memory access in terms of ICL trend. In contrast, the Streaming Transformer undergoes a significant performance drop once streaming begins at 6k steps. Although methods like StreamingLLM [31] address the issues faced by the Streaming Transformer, Memo achieves strong performance without requiring such modifications. Additionally, Streaming Memo provides a way to maintain peak ICL performance by identifying the highest-performing context length during evaluation and initiating streaming beyond that.

## 4.7 Finetuning

In Figures 2b and 4a, we see a saturation in Memo's ICL performance at ∼60% SR. To investigate whether this limitation stems from suboptimal context length extrapolation, we take Memo's and FCT's checkpoints trained until 1B steps and fine-tune them on a 4× larger context size of 16,384 tokens for 500M steps. As shown in Figure 4b, the ICL performance of both models improves when evaluated over 32k environment steps. Notably, Memo (16k) outperforms Memo (4k) in SPL, indicating that training with longer contexts enables the policy to find more efficient paths. While the 16k full-context transformer (FCT) improves significantly over its 4k variant, it still achieves lower maximum SR and SPL compared to Memo (16k). Interestingly, Memo (16k) generalizes up to 1.5× its training context length after which it sees a slight degradation, whereas FCT (16k) maintains performance up to at least 2×. This necessitates further research into the factors limiting Memo's long-context generalization, which we leave for future work.

## 4.8 Ablations.

**Summary Length Ablation**: There is a trade-off between creating or retaining more memory and the compute saved when generating fewer tokens at training or inference time. Additionally, creating more tokens pushes the model further into the "extrapolation" regime, where its positional encodings stop generalizing effectively. We vary the summary length between values 16, 32, and 64 (thereby varying the compression ratio, $l_{seg}/l_{sum}$, between 16×, 8×, and 4×) on the EXTOBJNAV task and plot the resulting performance in Figure 5a. We find that Memo is highly sensitive to the choice of summary length. The main performance gap emerges after 6k environment steps, where a summary length of 32 achieves the best generalization across longer trajectories. In contrast, a summary length of 64 performs the worst, likely due to summary tokens overfitting to the training context length (see Figure 10a), leading to redundant information storage and a lower signal-to-noise ratio.

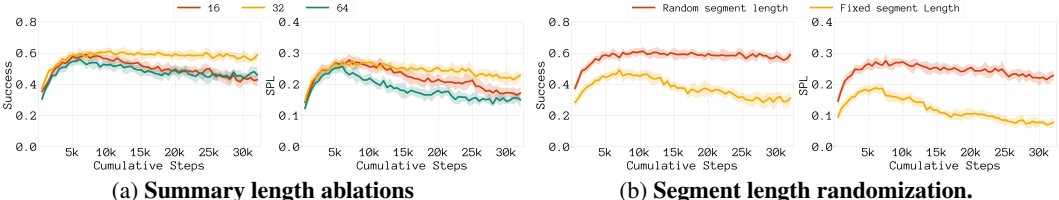

(a) **Summary length ablations**          (b) **Segment length randomization.**

Figure 5: **Memo ablations. (Left)** Effect of varying the number of memory tokens (16/32/64), where 32 outperforms 16, and 16 outperforms 64. **(Right)** Performance of Memo on EXTOBJNAV with and without randomization of the segment lengths at the end of which summarization is done, showing the latter is significantly less data-efficient.

**Segment Randomization**: In our main experiments, we adopted the strategy of randomizing segment lengths as in [7] for Memo, AC, and RMT to make token generation more robust to specific environment time steps. We randomized segment lengths between $[0.8, 1.2] \times 256$, resulting in a range of $[205, 307]$. In this ablation, we disable this randomization and evaluate the impact of using a fixed segment length of 256, as shown in Figure 5b. We observe that this variant performs significantly worse, not just during evaluation but in also training (see Figure 10b). We hypothesize that segment length randomization not only improves generalization by preventing overfitting to fixed boundaries but also serves as a form of curriculum. By exposing the model to segments of varying lengths, it naturally encounters both easier (shorter) and harder (longer) compression tasks, fostering a progressive learning effect. We present further results and ablations in Appendices A.6 and A.8.

## 5   Discussion

In this work, we proposed a simple yet effective approach for training transformers with memory in sequential decision-making tasks that require long-horizon reasoning. By introducing a context summarization mechanism, Memo enables transformers to retain and retrieve task-relevant information efficiently, significantly reducing computational overhead while maintaining long-term dependencies. Our experiments demonstrated Memo's key advantages over full-context transformer baselines. In terms of performance vs. efficiency, Memo achieves comparable or superior performance while attending to sequences over 8× shorter than the full-context transformer. This improves scalability, enhances extrapolation to longer contexts, and ensures robust performance in streaming settings, where past context must be discarded due to inference constraints.

Furthermore, our analysis highlighted two key design choices: propagating gradients across multiple summarization steps and accumulating summaries rather than using a fixed memory size. We showed that allowing gradients to flow through sequential memory updates significantly enhances long-horizon reasoning, while memory accumulation improves retention of relevant past experiences, leading to more effective decision-making.

## 6   Limitations

Our experiments focus on object navigation in unseen scenes, where agents locate a fixed set of randomly placed objects. We do not evaluate semantic generalization—i.e., navigating to entirely new object categories—since this would conflate memorization with general object recognition capability. Future work could explore leveraging foundation models in a more open-ended setting.

Additionally, while Memo effectively summarizes experience, we do not explore more flexible memory mechanisms, such as memory consolidation, where past summaries are progressively compressed. Investigating these strategies could further improve long-term memory efficiency. We also do not explicitly study length extrapolation. Our results indicate that compressed memory tokens improve robustness to longer contexts and enable streaming inference, but extending generalization beyond training limits remains an open challenge. Lastly, our memory mechanism is trained end-to-end via the RL objective. Future work could explore self-supervised objectives, such as future prediction, to enhance data efficiency in training memory representations.

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

# A   Appendix.

## A.1   Task Setting and Model architecture: EXTOBJNAV

Our policy takes as input an RGB image, the agent's relative position, the previous action, and the current goal object category. The RGB image is encoded using a ViT-B-sized VC-1 [20] visual encoder, finetuned by ReLIC to improve small object detection, a limitation of the original VC-1 model. The finetuned encoder remains frozen in all experiments, and we use VC-1's CLS token downstream, applying an MLP for dimension reduction to size 256. The goal category one-hot vector and previous action are also embedded into 256 dimensions, while the agent's position is converted into a 32-dimensional embedding. All these embeddings are concatenated together and passed to the sequence encoder.

All our sequence encoders (Memo, AC, etc) are based on a much smaller version of the causal auto-regressive transformer architecture used in LLaMA [29] (see details in Table 2). The weights of the model are trained from scratch unless specified otherwise. To accelerate data collection, the transformer's KV cache is maintained between rollout steps. Following ReLIC, we shuffle older episodes within each sequence and update the KV cache after each policy update, ensuring efficient memory reuse.

We use the same reward function for the navigation task as defined in Elawady et al. [11]. The reward consists of three components: (1) Geodesic distance reward: The agent receives a reward

$$r_d = -\Delta d$$

where $d$ is the geodesic distance to the closest object, which can change throughout the episode. (2) Slack penalty: A constant penalty of $-0.003$ per step encourages efficient navigation. (3) Success reward: The agent receives a reward of 2 upon successfully reaching the target.

The SPL (Success weighted by Path Length) metric is computed as:

$$\text{SPL} = \frac{1}{N} \sum_{i=1}^{N} S_i \cdot \frac{L_i^*}{\max(L_i, L_i^*)}$$

where $N$ is the number of episodes, $S_i$ is a binary indicator of success in episode $i$, $L_i$ is the length of the path taken by the agent, $L_i^*$ is the length of the shortest path to the goal (computed using privileged simulator information). The SPL values reported in the main graphs are computed at intervals of 500 environment steps. Specifically, the SPL at step $t$ is calculated by averaging the SPL of all episodes that completed between steps $t - 499$ and $t$. So the SPL at step 5000 reflects the performance of all episodes that finished between step 4501 and 5000.

## A.2   Task Setting and Model architecture: Dark-Key-To-Door

We refer the reader to the AMAGO implementation [12] (https://github.com/UT-Austin-RPL/amago) for details on the policy architecture used on the Dark-Key-To-Door task. We keep all implementation and training details fixed.

## A.3   Training Scenes Dataset: EXTOBJNAV

Since the training dataset used in ReLIC only consisted of 333 episodes, we create our own training dataset while using the same validation dataset as them. We use the same 37 train scenes as [11, 32]. However we generate 300 episodes per scene each having a different object placement. This leads to a total of 11100 training episode. In each episode we reduce the number of sampled objects to make the navigation task harder since the agent has to traverse longer, more targeted paths to reach objects instead of getting away by guessing (i.e. navigating to rooms with more clutter and many receptacles since the likelihood of finding an object would be higher there). We visualize the difference in sampled object distributions over episode, between the old [11] and new datasets in Figure 6.

## A.4   Pseudocode for Memo

We present the algorithmic pseudocode for Memo in Algorithms 1 and 2. The modifications with respect to a distributed PPO implementation (corresponding to ReLIC) are highlighted in blue. We plan to fully open-source our implementation upon publication and include an early release version at `https://github.com/Memory-icrl/memo`.

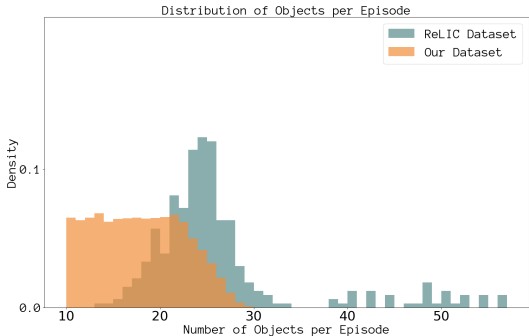

Figure 6: **Distribution of Objects in the training episodes:** We visualize the changed distribution of objects sampled (and placed) randomly in the scenes corresponding to our dataset versus the original training dataset used in [11]. In our training scenes, avoid sampling more than 30 objects per scene to avoid cluttered rooms where multiple target objects could be placed together, to reduce the risk of models overfitting during training.

---

**Algorithm 1** Memo
---

1: Initialize $\theta$, $\phi$, buffer $\mathcal{B} \leftarrow \emptyset$, summary tokens $\mathcal{S} \leftarrow \emptyset$
2: $o_1 \leftarrow \text{reset}()$
3: $i \leftarrow 0$
4: **for** $t = 1$ to $N$ **do**
5:     $\mathcal{C}_{i:t-1} \leftarrow$ context from $\mathcal{B}$
6:     $h_t \leftarrow SeqEnc_\phi(\mathcal{S}, \mathcal{C}_{i:t-1}, o_t)$
7:     $a \sim \pi_\theta(a \mid h_t)$
8:     $(r, o_{t+1}, done) \leftarrow \text{step}(a)$
9:     **if** $done$ **then**
10:         $o_{t+1} \leftarrow \text{reset}()$
11:     **end if**
12:     **if** $t \ \% \ l_{seg} = 0$ **then**                                      $\triangleright l_{seg}$ is segment length
13:         $\mathcal{S} \leftarrow SeqEnc_\phi(\mathcal{S}, \mathcal{C}_{i:t-1}, o_t)$
14:         $i \leftarrow t$
15:     **end if**
16:     Append $(o_t, a, r, o_{t+1}, done)$ to $\mathcal{B}$
17:     **if** $t \ \% \ \text{num\_steps} = 0$ **then**
18:         $\text{TRAIN}(\mathcal{B}, t)$
19:     **end if**
20:     $o_t \leftarrow o_{t+1}$
21: **end for**

---

**Algorithm 2** Train Memo
---

1: **procedure** $\text{TRAIN}(\mathcal{B}, t)$
2:     Initialize summary tokens $\mathcal{S}_{train} \leftarrow \emptyset$
3:     $n \leftarrow \left\lfloor \frac{t}{l_{seg}} \right\rfloor$                                          $\triangleright$ n is num segments
4:     **for** $j = 0$ to n **do**
5:         $\mathcal{C}_{jl_{seg}:(j+1)l_{seg}} \leftarrow$ context from $\mathcal{B}$
6:         $\mathcal{S}_{train} \leftarrow SeqEnc_\phi(\mathcal{S}_{train}, \mathcal{C}_{jl_{seg}:(j+1)l_{seg}})$
7:     **end for**
8:     $\mathcal{C}_{nl_{seg}:t} \leftarrow$ context from $\mathcal{B}$
9:     $\mathcal{L} \leftarrow PPO_{loss}(\mathcal{S}_{train}, \mathcal{C}_{nl_{seg}:t})$
10:     $\theta \leftarrow \theta - \eta \nabla_\theta \mathcal{L}$
11: **end procedure**

## A.5   Extended Result: FLOPs Comparison

Here, we include memory, FLOPs and latency comparison between Memo and Full-Context Transformer during evaluation on EXTOBJNAV at 32k env steps. Memo requires 10x lesser memory and 4.2x lesser floating point operation, being 2x faster than the Full-Context Transformer

Table 1: **GPU memory usage comparison**: Memo versus the Full-Context Transformer (FCT) at the end of 32k steps of evaluation on the EXTOBJNAV task. We observe a $10\times$ higher memory requirement for the KV cache of FCT, in line with the context compression ratio ($\sim 8$) of Memo.

|  | **Memo** | **Full-Context Transformer** |
|---|---|---|
| GPU Memory | 51.8 MB | 546.5 MB |
| Model FLOPs | 17.61 MFLOPs | 74.49 MFLOPs |
| Latency | 5.3 ms | 10.1 ms |

## A.6   Extended Result: Memory-Maze Benchmark

To further evaluate our method on long-horizon memory-intensive tasks, we include preliminary results on the Memory Maze benchmark. This environment poses a challenging multi-goal navigation task in a procedurally generated Mujoco-based maze. At the start of each episode, up to six colored goal objects are randomly placed in the maze, and the agent is given a randomly sampled sequence of goals to reach in order. Upon reaching the current goal, the next target is revealed, continuing until the episode ends. We evaluate Memo and the Full-Context Transformer (FCT) on the 9×9 version of the task, which includes two obstacles and has an episode length of 1000 steps. This setting demands both exploration and the ability to recall spatial information about previously encountered goals and maze structure. Figure 8 show that Memo matches the performance of FCT on this task.

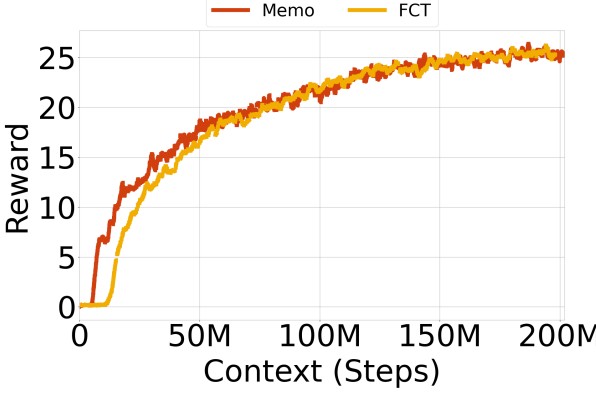

Figure 7: Returns of FCT vs AC on Memory Maze 9x9.

## A.7   Extended Result: Benefits of Memory Accumulation on an Adversarially Long-Context RL Task (T-Maze)

To explore memory-reuse, recall that we had included a recurrent-only summarisation (RMT) baseline in our experiments in Section 4.4. While RMT performance improved with larger memory sizes (e.g., RMT-64 outperformed RMT-32), Memo still achieved $\tilde{5}$% higher success rate on navigation. Runs with a larger summary size (RMT-128) showed unstable convergence.

We observed an interesting trend across recurrent summarization baseline results on different benchmarks: while recurrent summarization can eventually achieve reasonable performance, they typically require significantly more training due to the need to propagate gradients through many sequential memory updates. In contrast, architectures like Memo—and transformers more generally—enable more efficient credit assignment by allowing earlier memories to receive gradient signals from later timesteps via direct attention mechanisms.To further validate the above intuition, we ran an experiment in the synthetic T-maze gridworld environment [22]. Here, the agent sees a clue ("left" or

"right") at timestep 0, which disappears at timestep 1. It then traverses a long corridor with only forward actions, and at the end must choose the left or right room based on the initial clue to receive a reward. The corridor length can be made arbitrarily large (e.g., 10,000 steps). We hypothesized that purely recurrent models would especially struggle in this adversarial setting, as learning to retain the clue requires backpropagating gradients through all the segments corresponding to the 10,000 steps. Memo and FCT, however, can access that information directly at later timesteps—either through full attention (FCT) or through a small number of memory consolidation stages (Memo). Empirically, we observed the expected result: RMT required 10× longer to achieve an average reward of 1.0 for the first time during training, compared to FCT and Memo. In addition, it exhibited much greater instability in learning the correct policy, since it did not manage to maintain this performance (average reward = 1.0) for three consecutive checkpoints at any point during training.

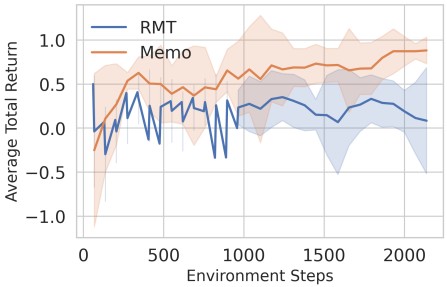

Figure 8: Returns of Memo-RMT vs Memo on T-Maze ($l = 2000$).

## A.8 Extended Ablation: Direct KV Cache Construction from Summary Embeddings

To assess whether the improved performance of Memo stems from the additional computation it uses (due to summary embeddings being processed through the transformer twice), we perform an ablation that removes the re-encoding step.

In this variant, instead of generating summary tokens and re-encoding them through the transformer, we directly construct the memory (KV cache) from the internal embeddings produced at each layer while encoding the learnable summary embeddings. As in Memo, these summary embeddings are inserted at regular intervals between segments of tokens. However, unlike Memo, their layer-wise representations are extracted directly to serve as the KV cache, without further processing.

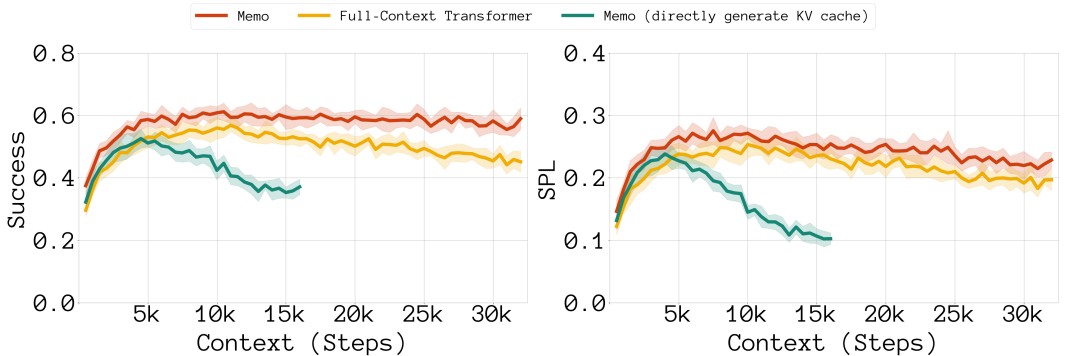

Figure 9: **Ablating the summary generation process in Memo**.

This approach is more compute-efficient, as each token—including summaries—is passed through the transformer only once. However, as shown in Figure 9, this variant performs poorly and quickly degrades beyond the training context length. We hypothesize that this is not only due to the lack of deeper summarization capacity, but also due to positional embedding generalization issues. Since summary embeddings are appended at the end of each segment, their positional indices lie in the range $[l_{seg}, l_{seg} + l_{sum}]$, with offsets accumulating across segments. Consequently, these summary positions - and those of all future tokens - progressively reach larger values, potentially leading to distribution shifts in the positional encodings that the model is unable to generalize over.

This result suggests that Memo's re-encoding step not only adds depth for more expressive summarization but also resets the positional index range for the summary tokens, both of which appear beneficial for stability and generalization.

## A.9 Training-Validation Gap

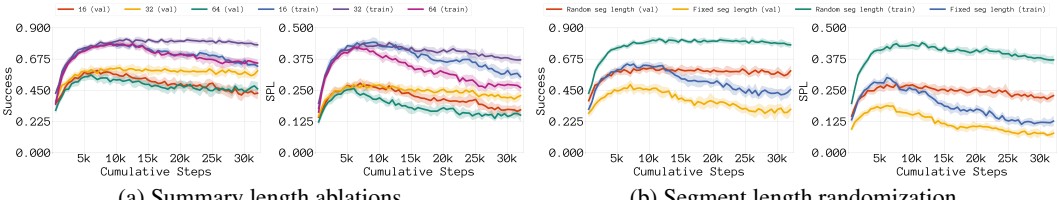

| (a) Summary length ablations. | (b) Segment length randomization. |

Figure 10: **Training-validation gap for Memo ablations on EXTOBJNAV. (Left)** All summary tokens sizes (16/32/64) perform equally well on the training set for $\sim 7.5k$ environment steps after which 16 and 64 summary token policies start degrading faster. **(Right)** Fixed segment length's training performance is worse than the val performance of random segment length.

In this section, we visualize the overlaid training and validation performance curves for Memo on the EXTOBJNAV task, which presents two key challenges. First, validation scenes are entirely novel, requiring models to generalize their in-context learning (ICL) solutions. Second, models are evaluated on 4× larger trial sizes, testing their context extrapolation capabilities. To assess generalization, we compare overfitting levels across models, distinguishing whether low validation performance stems from limited encoding expressivity (indicated by low training performance) or poor generalization despite strong training performance. We present train versus validation curves for Memo while ablating the summary lengths in Figure 10a and while ablating the segment length randomization in Figure 10b.

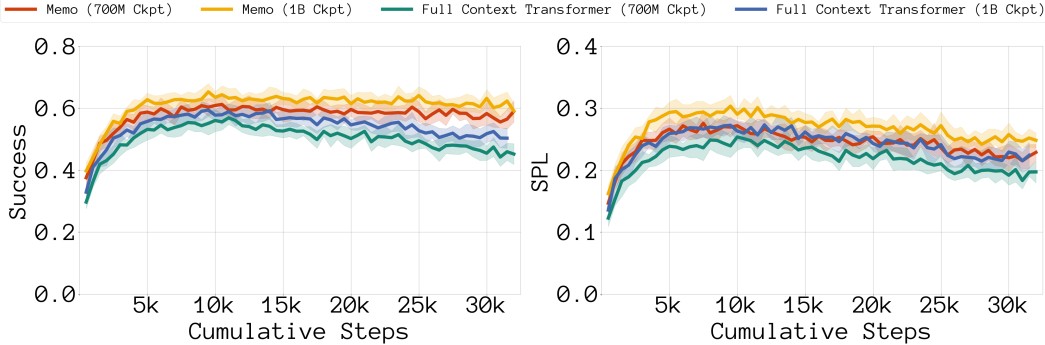

Figure 11: We compare the validation reward curves of checkpoints saved at 700M and 1000M steps of training for Memo and the full context transformer (FCT) variants on the EXTOBJNAV task trained with ReLIC. Both checkpoints of Memo acheve higher success rate compared to the checkpoints of FCT, while its 1000M approaches the 700M Memo checkpoint's performance in terms of the SPL metric. This highlights the sample efficiency of Memo, besides the compute efficiency which comes from using much smaller contexts in the transformer encoding and decoding process.

## A.10 Sample Efficiency of Memo

We observed higher sample-efficiency for Memo during training on the EXTOBJNAV and Dark-Key-To-Door tasks compared to the full context transformer. For Dark-Key-To-Door, this is directly observable in Figure 3a since the performance is plotted against training progress. We present the this observation for the EXTOBJNAV task in this section by visualizing the performance of earlier and later checkpoints (700M and 1B training steps) of both FCT and Memo in Appendix A.9. We see that even earlier into training, Memo exhibits higher-ICL success rates compared to FCT which starts to get closer to Memo's SPL only after 300M more steps of training.

### A.11 Hyperparameters

| Hyperparameter | Ours/AC/RMT | Transformer |
|---|---|---|
| **Model Architecture** | | |
| # Layers | | 4 |
| # Heads | | 8 |
| Hidden dimensions | | 256 |
| MLP Hidden dimensions | | 1024 |
| Activation | | GeLU [26] |
| # Sink-KV | 0 | 1 |
| Attention sink | N.A. | Sink $KV_0$ |
| Episode index encoding | N.A. | RoPE [28] |
| Within-episode position encoding | RoPE [28] | Learnable |
| **Training Setup** | | |
| Workers | 320 (20 env workers per GPU $\times$ 16 GPUs) | |
| Batch Size | 160 | |
| RL Algorithm | DDPPO [30] | |
| Discount Factor ($\gamma$) | 0.99 | |
| GAE Parameter ($\tau$) | 0.95 | |
| Entropy Coefficient | 0.1 | |
| Value Loss Coefficient | 0.5 | |
| **Optimization** | | |
| Optimizer | Adam [17] | |
| Regularization | Depth dropout [15] with value 0.1 | |
| | In-context episodes shuffled after partial updates | |
| Learning Rate Schedule | Warm-up for 100K env interactions | |
| Initial Learning Rate | $4 \times 10^{-7}$ | |
| Learning Rate at Warm-up End | $4 \times 10^{-4}$ | |
| Decay Schedule | Cosine decay [19] to 0 after 1B steps | |
| **Computation** | | |
| Precision | FP16 for visual encoder, FP32 for other model components | |
| Rollout size | 4096 | |
| Total # updates per rollout | 16 | |
| # partial updates | 15 | |
| # full updates | 1 | |
| **Hardware and Performance** | | |
| Environment Steps | 700M Steps | |
| Hardware | 16x NVIDIA A40 GPUs | |
| Training Time | 2.5 days | |

Table 2: Comprehensive training setup and hyperparameters related to the EXTOBJNAV task.

