# OpenReview forum: "Memo: Training Memory-Efficient Embodied Agents with Reinforcement Learning"
_NeurIPS.cc/2025/Conference — NeurIPS 2025 spotlight_

### Official Review · Reviewer_VTDX · 2025-07-02

**Clarity:** 2
**Significance:** 3
**Originality:** 3
**Rating:** 4
**Confidence:** 4

**Summary:**

In this paper, the authors propose Memo, which is a transformer-based RL framework that uses periodic summary tokens to compress historical context, enabling efficient long-horizon decision-making. Memo interleaves learnable summarization tokens at fixed intervals during training, creating a condensed memory buffer that replaces the full KV cache. Experimental evaluations on ExtObjNav (3D navigation) and Dark-Key-To-Door, show that Memo outperforms full-context transformers (FCT) and recurrent baselines (RMT).

**Questions:**

1. Why does Memo generalize worse than FCT beyond 1.5× context (Fig. 4b)? Is this due to positional encoding saturation or information loss in summaries? Provide analysis beyond speculation.

2. How does Memo ensure summaries capture task-critical information (e.g., object locations vs. irrelevant details)? The RL objective may not suffice for optimal compression.

3. Memo requires 16×A40 GPUs for 2.5 days. Does the inference efficiency justify this? Compare total energy/compute vs. FCT.

4. When does Memo perform poorly? E.g., in tasks requiring high-fidelity recall (e.g., "wall color as landmark" in Sec. 1), could summarization discard crucial details?

**Ethical Concerns:**

["NO or VERY MINOR ethics concerns only"]

**Limitations:**

N/a.

**Paper Formatting Concerns:**

N/a.

**Quality:**

2

**Strengths And Weaknesses:**

## Pros
1. The topic of this paper is critical and very interesting. Memo adapts language-model context compression to RL’s unique challenges (credit assignment, partial observability), filling a gap in memory-efficient embodied AI.

2. Extensive experiments across tasks (grid-world, photorealistic sim), RL algorithms (on/off-policy), and metrics (SR, SPL, FLOPs, latency) show the effectiveness of MeMo.

3. Memo extrapolates better to longer contexts than FCT (Fig. 4b) and shows stable convergence where FCT fails (Fig. 3a).

4. Well-designed attention masking, positional encoding, and KV-cache refresh mechanisms ensure gradient flow and memory consistency.

## Cons

1. In this paper, the experiments focus mainly on navigation with fixed object sets. No evaluation of semantic generalization (e.g., novel object categories) or dynamic environments, raising questions about scalability to real-world embodied tasks.

2. Memo degrades beyond 1.5× training context (vs. FCT’s 2×). The paper attributes this to "suboptimal extrapolation" but lacks investigation or explanation into root causes (e.g., positional encoding drift).

3. The paper omits some comparisons/ablations to state-of-the-art memory methods (e.g., Memorizing Transformers, StreamingLLM) or hybrid architectures (e.g., transformer-RNN).

---

> ### Author Rebuttal · Authors · 2025-07-31
>
> We thank the reviewer for their positive feedback and recognition of our work, including the comprehensive experiments and the critical problem addressed. We also appreciate the acknowledgment of Memo’s superior stability in performance and convergence over longer contexts.
>
> We address the remaining concerns and questions below, and we remain open to discussing any further points during the discussion period.
>
> > In this paper, the experiments focus mainly on navigation with fixed object sets. No evaluation of semantic generalization (e.g., novel object categories) or dynamic environments, raising questions about scalability to real-world embodied tasks.
>
> We agree. However, at the time of our work, no established benchmarks offered both the diversity and long-range dependencies necessary to meaningfully train or evaluate such capabilities in embodied agents. That said, our method is designed with these challenges in mind and its general context summarization mechanism should be trainable given any long horizon task. We see the development of benchmarks targeting long-horizon memory in open-ended settings as a key enabler for future work, and we plan to pursue this in follow-up research.
>
>
> > Why does Memo generalize worse than FCT beyond 1.5× context (Fig. 4b)? Is this due to positional encoding saturation or information loss in summaries? Provide analysis beyond speculation.
>
> Great question. We conducted several targeted evaluations to better understand this behavior.
>
> First, we note that Memo does generalize well under a streaming evaluation regime. In Fig. 4a (trained on 4k), Memo performs competitively with FCT even when past summaries are replaced continually, suggesting that **older summaries can be replaced by newer ones without degradation—as long as the model operates within the distribution of input lengths it was trained on**.
> To further probe the 16k-finetuned Memo model used in Fig. 4b, we ran two streaming evaluations:
>
> - **Recent Memories Only:** We retained only memory summaries corresponding to the most recent 20k environment steps and discarded all earlier memories.
> - **Old + Recent:** We kept only the oldest 20k memory steps and appended a single recent memory chunk (assigning it position index N+1, where N is the number of old memories).
>
> In both cases, Memo maintained its performance after streaming began. This suggests that information loss in summaries is not the primary cause of degradation when generalizing beyond 1.5× context.
>
> The degradation likely stems from a distributional shift at inference time—specifically, Memo attending over far more input tokens than it was trained on. Due to summarization, Memo sees only a bounded number of memory tokens during training, whereas FCT is consistently exposed to full-length sequences. This also limits the range of positional embeddings that Memo experiences, further impacting its ability to generalize to longer contexts. These observations are consistent with findings in [1], which show that transformers trained on longer sequences generalize better to even longer ones. We believe Memo’s quicker degradation is a consequence of both limited positional exposure and reduced effective context length during training.
>
> [1] Zhou et. al, Transformers Can Achieve Length Generalization But Not Robustly
>
> > The paper omits some comparisons/ablations to state-of-the-art memory methods (e.g., Memorizing Transformers, StreamingLLM) or hybrid architectures (e.g., transformer-RNN).
>
> **StreamingLLM:**
>
> In the paper, we already incorporate two aspects inspired by StreamingLLM.
>
> - The FCT baseline we used (from the RELIC paper) already includes an attention sink token, as recommended by StreamingLLM. RELIC had ablated this and found the sink token version to perform better.
>
> - We also included evaluations in a streaming setup where we do windowed inference over the KV cache, and found that FCT’s performance degraded significantly—even with a sink token (Figure 4a). We explored several mitigation strategies inspired by StreamingLLM (e.g., retaining a few early tokens along with the most recent window of tokens, preserving the tokens for the first episode in full to avoid mid-episode truncation discontinuities etc), but none resolved the degradation. In contrast, Memo used a simple FIFO strategy for selecting memory tokens and remained robust without any special design changes.
>
> **Mamba:**
>
> Following the suggestion from SD9F, we evaluated the Mamba architecture on the Dark-Key-To-Door task. Mamba achieved an average reward of approximately 43, compared to 55 for both FCT and Memo, and required about twice the training time to converge. This is consistent with a broader trend we observed across recurrent baselines: **while recurrent models can eventually achieve reasonable performance, they typically require significantly more training due to the need to propagate gradients through many sequential memory updates**. In contrast, architectures like Memo—and transformers more generally—enable more efficient credit assignment by allowing earlier memories to receive gradient signals from later timesteps via direct attention mechanisms. We will add a similar experiment for Mamba on the extended object navigation task in the revised manuscript.
>
> **Transformer-XL (Tr-XL):**
> We also evaluated Tr-XL to test approaches where memory creation is based on fixed heuristics and does not benefit directly from learning signals (e.g., using a frozen kv cache or averaging the cache elements). As also reported in RELIC [3], **Tr-XL performed poorly in our setting**, achieving only ~15% success and showing no in-context learning capabilities.
>
> > How does Memo ensure that summaries capture task-critical information (e.g., object locations vs. irrelevant details)? The RL objective alone may not guarantee optimal compression.
>
> Memo is trained end-to-end using task-specific reinforcement learning objectives, such as navigation success. As a result, the summarization mechanism is implicitly optimized to retain information that directly impacts downstream performance (e.g., object locations, path-finding cues), while discarding irrelevant or redundant details (e.g., wall colors, if they are not task-relevant). We emphasize that this form of lossy compression is not arbitrary; rather, it is shaped by the learning signal provided through interaction. If certain features consistently influence task success, the model learns to preserve them through the summary mechanism. This stands in contrast to manually designed memory systems that may enforce fixed heuristics, which can be suboptimal across diverse tasks.
>
> That said, it would be interesting to explore self supervised learning for pre-training the memorization, especially when considering large vision-language models and multi-modal dialogue datasets which implicitly capture what information humans tend to be interested in and talk about.
>
> > When does Memo perform poorly? Could summarization discard crucial details in tasks requiring high-fidelity recall (e.g., "wall color as landmark" in Sec. 1)?
>
> We note that the failure modes we observe in Memo are qualitatively similar to those seen in baseline methods (e.g., agents getting stuck, or failing to detect small objects from a distance). This suggests that the summarization mechanism is likely not introducing new or unique failure modes, but rather inherits the broader limitations faced by embodied agents in complex visual environments. Addressing these foundational challenges in perception and exploration remains an important direction for future work.
>
> > Memo requires 16×A40 GPUs for 2.5 days. Does the inference efficiency justify this? Compare total energy/compute vs. FCT.
>
> Memo only takes 15% longer to train compared to FCT while being substantially more efficient during inference.
> - For a 4k sequence length training setup, FCT training takes ~60 hours with Memo taking 69.5 hours. The additional time in Memo stems from its segment-wise processing during PPO updates, which incurs extra forward passes on a split-up context.
> - Memory usage per GPU during training is also comparable: FCT uses ~48.3% of GPU memory per device, while Memo uses ~46.9% (we weren’t able to double the batch size as some GPUs exceeded 50% memory usage).
> - During inference at 32k environment steps, **Memo is 2× faster (5.3 ms vs 10.1 ms per step) than FCT** and **Memo uses 10× less GPU memory (51.8 MB vs 546.5 MB)**.
>
> This significant reduction in inference cost is especially **valuable in embodied AI** scenarios, where models are deployed on edge devices with strict memory and latency constraints.  In such settings, multiple models—e.g., perception, mapping, and policy—often run concurrently on a single robot. A smaller and faster policy model, like Memo, frees up memory and compute, allowing other components to operate more efficiently or enabling the deployment of more capable models within the same hardware.
>
> Thanks again for your constructive and encouraging feedback!

---

> > ### Comment · Reviewer_VTDX · 2025-08-03
> > **Thanks**
> >
> > Thank you for the thorough and detailed rebuttal, so I maintain my positive score.

---

### Official Review · Reviewer_Q5LN · 2025-07-02

**Clarity:** 3
**Significance:** 2
**Originality:** 2
**Rating:** 4
**Confidence:** 3

**Summary:**

This paper introduces a transformer-based architecture and training method for reinforcement learning (RL) on memory-intensive, long-horizon tasks. The proposed model, Memo, summarizes and stores task-relevant past experiences using specialized summary tokens, which are fed back into the model at future timesteps. Crucially, Memo integrates this summarization mechanism directly into the RL optimization process, allowing gradients to propagate across all summaries by attending to and refining them. The method is evaluated on a grid-world meta-RL benchmark and a multi-object navigation (ExtObjNav) task.
Memo addresses a key challenge in RL: unlike standard transformers that assume access to the full past context, RL agents must selectively retain and access task-relevant memories aligned with the reward signal. The proposed approach outperforms transformer-based baselines—despite having access to fewer tokens—in both scalability and efficiency.

**Questions:**

Please see the weaknesses.

**Ethical Concerns:**

["NO or VERY MINOR ethics concerns only"]

**Final Justification:**

I keep my positive score. Most of my concerns were addressed. The authors explained about the training overhead of propagating gradients to summary tokens, they mentioned an error in the RMT plot which will be fixed for the camera-ready with more expected results, and they explained the novelty of the method compared to Autocompressors (AC).

**Limitations:**

yes

**Paper Formatting Concerns:**

No concerns.

**Quality:**

3

**Strengths And Weaknesses:**

Strengths:
- The paper includes comprehensive experiments, strong baselines, and thorough ablations to demonstrate the effectiveness of Memo.
- The paper demonstrates that Memo improves performance across both on-policy and off-policy RL settings.
- It compares Memo to several strong baselines: full-context transformers, transformers without in-context learning, and recurrent memory transformers with fixed-size memory constraints.
- On the ExtObjNav task, Memo achieves a 7.5% higher success rate and better in-context learning compared to the full-context transformer, while using 8x fewer in-context tokens and achieving 2x lower inference latency.
- While both Memo and the full-context transformer show improved performance up to 10k steps in ExtObjNav, the full-context transformer’s performance degrades afterward—whereas Memo maintains higher performance—indicating more stable and effective in-context learning.

Weaknesses:
- Does the ability to propagate gradients to all past summary tokens introduce significant training overhead? How does this affect training time compared to baselines?
- RMT appears unstable when using a higher number of summary tokens, especially in ExtObjNav. Could the authors clarify why this degradation occurs only in this task and not in the Dark-Key-to-Door task?
- How does Memo differ architecturally and methodologically from Autocompressors (AC)? Is the novelty primarily in applying the summarization approach to RL and training from scratch?
- Memo is highly sensitive to the number of memory tokens, with performance degrading significantly when increasing the token count to 64. Can the authors provide intuition for this sensitivity?

---

> ### Author Rebuttal · Authors · 2025-07-31
>
> We thank the reviewer for their constructive feedback and appreciate their positive reception of our work, particularly for recognizing our experiment as comprehensive and well-designed with thoughtful ablations.
>
> We address the remaining concerns and questions below, and we remain open to discussing any further points during the discussion period.
>
> > Does the ability to propagate gradients to all past summary tokens introduce significant training overhead? How does this affect training time compared to baselines?
>
> - In terms of wallclock time, it takes 60 vs 69 hours for our training runs of FCT (Full Context Transformer) v/s Memo (due to forward passes on sequence of split up context instead of one forward pass on a longer context). In contrast it takes 57 hrs for the same number of training steps for AC (TBTT).
> - In practice, we observe that there aren’t significant differences in the train time memory GPU usage of Memo v/s FCT since Flash Attention already optimises GPU memory to be below what the naive implementation would take, and so the savings in Memo from much shallower attention operations (because now tokens don’t attend over very long contexts and instead over memories and immediate context) is somewhat cancelled out by the memory needed due to the extra KV cache corresponding to memory tokens.
> - While Memo requires a modest increase in training time, this is **compensated by substantial efficiency and performance gains**. Specifically, Memo is 2× faster than FCT at inference (5.3 ms vs. 10.1 ms per step) and consumes 10× less GPU memory (51.8 MB vs. 546.5 MB). Additionally, Memo outperforms the AC baseline in downstream task performance, achieving 62% success compared to 50% for AC.
>
> > RMT appears unstable when using a higher number of summary tokens, especially in ExtObjNav. Could the authors clarify why this degradation occurs only in this task and not in the Dark-Key-to-Door task?
>
> We had an important correction to make for the RMT plot: we reported the instability in the paper and after paper submission we decided to dig deeper into sources of bugs for that run and found an erroneous gradient accumulation config setting to be the cause of anomalous performance trends there. We reran and got much more expected results, the performance of the 64 summary len run is 4-5% higher than rmt-32 and 5% lower than the highest ICL performance achieved by Memo.  We will include this revised plot in the manuscript.
>
> > Memo is highly sensitive to the number of memory tokens, with performance degrading significantly when increasing the token count to 64. Can the authors provide intuition for this sensitivity?
>
> On both the training (Fig. 9a) and validation sets (Fig. 5a), Memo's performance is comparable across all three summary lengths up to ~6k context length. However, beyond this point, the configuration with 64 summary tokens exhibits the weakest generalization. While we do not yet have a definitive answer, we offer the following context and hypotheses that we plan to investigate further for the camera-ready version:
>
> - Consistency with observation in AC: The AC paper reports a similar sensitivity to summary length but does not provide a concrete explanation for this behavior, suggesting this is a broader phenomenon not unique to our setup.
> - Context Explosion and Extrapolation: Increasing the number of summary tokens may lead to faster context growth across segments, pushing the model into a harder extrapolation regime more quickly. This might explain why the 64-token setup degrades earlier than 32 and 16.
> - Training Exposure Gap: However, this doesn't fully explain why the 16-token run underperforms relative to the 32-token run. One possibility is that with only 16 summary tokens per segment, the model sees a much smaller context (and therefore range of positional embeddings) during training. If we follow the findings in [1], which show that transformers trained on long sequences generalize disproportionately better to even longer sequences compared to transformers trained on short sequences and then evaluated on sequences with the same extrapolation factor. We believe Memo(16)’s quicker degradation here could be a consequence of both limited positional exposure and reduced effective context length during training.
>
>
> Understanding this interaction fully would require targeted ablations, which we plan to explore by the camera-ready deadline.
>
>
> [1] Zhou et. al, Transformers Can Achieve Length Generalization But Not Robustly
>
>
> > How does Memo differ architecturally and methodologically from Autocompressors (AC)? Is the novelty primarily in applying the summarization approach to RL and training from scratch?
>
> Besides the differences mentioned by the reviewer, we further note some subtle differences in our experimental setup and findings below:
> - AC compares models based on average perplexity on later chunks in a sequence, using a token-matched baseline (e.g., comparing 50 summary tokens against 50–150 raw tokens, rather than the full 750-token context that 50 tokens were meant to be summarising). Our baseline (Full Context Transformer) has access to the entire raw context that Memo is meant to be summarizing—making the comparison more stringent. Unlike the straightforward addition of summary tokens during supervised training in AC, Memo operates in an online in-context RL setting, which brings in additional challenges: needing to manage a KV cache of both the recent history and memory tokens during rollouts in the rollout buffer, recomputing all encodings to maintain on-policy-ness during update steps, followed by refreshing the cache for subsequent data collection. Getting memo to work in this multi-episode RL setting required non-trivial effort, which we believe will now be easier for others to adopt with our open-sourced implementation (Appendix.4).
> - We show that Memo matches or outperforms the full-context baseline, even when streaming (i.e., pruning older summaries’ KV cache), whereas the full-context model degrades.
> - Finally, we show that end-to-end trainability of the context is critical: Memo allows gradients to flow across the entire context, whereas TBTT (Truncated Backpropagation Through Time), as used in AC, fails to capture long-term dependencies effectively. This result has wider implications when trying to adopt AC for training LLMs to do long context tasks that impose a different reasoning problem from what a short context model has learnt to do.
>
> Thanks again for your constructive and encouraging feedback!

---

> ### Author Response · Authors · 2025-08-06
>
> Thank you again for your detailed review and thoughtful feedback. We just wanted to check in to see if there are any remaining concerns or questions we can help clarify during the discussion period.

---

> > ### Comment · Reviewer_Q5LN · 2025-08-06
> >
> > Thank you for your detailed rebuttal. Most of my concerns are clarified. I keep my positive score.

---

### Official Review · Reviewer_SD9F · 2025-07-03

**Clarity:** 3
**Significance:** 3
**Originality:** 2
**Rating:** 5
**Confidence:** 4

**Summary:**

The paper shows how to add memory tokens to a simple model-free Transformer-based RL pipeline, making it more compute and storage efficient, while outperforming baselines.

**Questions:**

* Did you consider using learnable positional encodings for the memory tokens?

* If we reuse memory tokens rather than adding new ones each time, does the performance degrade significantly?

**Ethical Concerns:**

["NO or VERY MINOR ethics concerns only"]

**Final Justification:**

I think the paper is technically strong and novel, and I am happy with the rebuttal and the additional experiments.

**Limitations:**

-

**Paper Formatting Concerns:**

-

**Quality:**

3

**Strengths And Weaknesses:**

I found the paper straightforward and addressing an important problem, with well-designed experiments, strong results, and a thoughtful ablation study. The writing is clear. The results look good and largely as expected.

**Weaknesses**

The most concerning aspect for me is the absence of comparisons with RNN-, SSM-, or Mamba-based baselines. Recurrent architectures eliminate the need to manage memory tokens or worry about context length, potentially making the proposed approach obsolete. So it would be valuable to include such comparisons. The authors mention that RL^2 doesn’t work well, which explains its exclusion. But what about SSM or Mamba layers? While I’m not an expert in applying SSM layers to RL, and I’m unaware of strong existing baselines using them, I understand that implementing and tuning them may be difficult and time-consuming. Nonetheless, I believe including such baselines would significantly strengthen the paper.

**Comments**

1) The approach may lack originality in the sense that it straightforwardly imports concepts from NLP into RL. However, I believe doing so is important, and I recognize that making such methods work reliably and perform well is non-trivial. This line of work is valuable for advancing in-context learning in RL.

2) It seems the authors report the worst AC results in Figure 2(b). I would recommend showing the best results (i.e., without truncated backpropagation).

---

> ### Author Rebuttal · Authors · 2025-07-31
>
> We thank the reviewer for their constructive feedback and positive reception of our work. We appreciate the recognition that our paper addresses a critical and interesting problem through clear writing and straightforward presentation.
>
> Below, we respond to the remaining concerns and questions raised. We remain open and happy to address any further points during the discussion period.
>
> >“Absence of comparisons with RNN-, SSM-, or Mamba-based baselines. “
>
> Thanks for the suggestion!
>
> We evaluated the **Mamba architecture on the Dark-Key-To-Door task**. Mamba achieved an average reward of approximately 43, compared to 55 for both FCT (Full Context Transformer) and Memo, and required about twice the training time to converge. This is consistent with a broader trend we observed across recurrent baselines: while **recurrent models can eventually achieve reasonable performance, they typically require significantly more training due to the need to propagate gradients through many sequential memory updates**. In contrast, architectures like Memo—and transformers more generally—enable more efficient credit assignment by allowing earlier memories to receive gradient signals from later timesteps via direct attention mechanisms.
>
> We will add a similar experiment for Mamba on the extended object navigation task in the revised manuscript.
>
>
> **Synthetic Validation in T-Maze:**
>
> - To further validate the above intuition, we ran an experiment in the synthetic T-maze gridworld environment [1]. Here, the agent sees a clue ("left" or "right") at timestep 0, which disappears at timestep 1. It then traverses a long corridor with only forward actions, and at the end must choose the left or right room based on the initial clue to receive a reward. The corridor length can be made arbitrarily large (e.g., 10,000 steps).
>
> - We hypothesized that purely recurrent models would struggle in this setting, as learning to retain the clue requires backpropagating gradients through all 10,000 steps. Memo and FCT, however, can access that information directly at later timesteps—either through full attention (FCT) or through a small number of memory consolidation stages (Memo). Empirically, we observed the expected result: **RMT required ~10× longer to converge and exhibited greater instability in learning the correct policy compared to FCT and Memo**.
>
> [1] Ni et. al, When do transformers shine in rl? decoupling memory from credit assignment
>
>
> > It seems the authors report the worst AC results in Figure 2(b). I would recommend showing the best results (i.e., without truncated backpropagation).
>
> In Fig 2(b), we tried to keep the setting close to the original AC implementation. The original AC method did not employ full backpropagation through all segments during fine-tuning, explicitly demonstrating that truncated backpropagation sufficed for their context-compression setting which had only 3 segments of summarisation on relatively short-range textual tasks. We are happy to replace the full-backpropagation variant in the main results figure if recommended by the reviewer.
>
>
> > Did you consider using learnable positional encodings for the memory tokens?
>
> We opted against learnable positional encodings because fixed encodings (like RoPE) generalize better when extrapolating to longer contexts at inference. Although learnable encodings could potentially help differentiate summary from observation tokens, they typically struggle with extrapolation, which is critical in our setting since our evaluation steps are 8x the number of training steps.
>
>
> > If we reuse memory tokens rather than adding new ones each time, does the performance degrade significantly?
>
> We appreciate this suggestion but would like to ensure we fully understand the intended modification. Could you please clarify what you mean by "reusing memory tokens" – would that mean repeatedly overwriting the same fixed set of summary tokens rather than introducing new tokens after each summarization segment (similar to RMT perhaps)? Additionally, could you elaborate on your motivation behind exploring this change—are you interested in memory efficiency, or another ablation?
>
>
> Thanks again for your positive and encouraging feedback!

---

> > ### Comment · Reviewer_SD9F · 2025-08-04
> >
> > Thank you for the response and the new Mamba experiments.
> >
> >  >  would that mean repeatedly overwriting the same fixed set of summary tokens rather than introducing new tokens after each summarization segment (similar to RMT perhaps)
> >
> > Yes, I'm referring to the RMT approach of reusing memory tokens.
> >
> > > Additionally, could you elaborate on your motivation behind exploring this change—are you interested in memory efficiency, or another ablation?
> >
> > Yes. I believe that constantly expanding memory becomes awkward and inefficient once a system is continual and reaches a certain scale. I don’t consider this a limitation of the current work, I’m simply curious about the potential downsides of adopting a more memory-efficient strategy.

---

> > > ### Author Response · Authors · 2025-08-06
> > >
> > > Thank you for the clarification and for raising this important point regarding memory reuse and scalability in continual settings.
> > >
> > > To explore memory-reuse, we included a RMT baseline in our experiments. While RMT performance improved with larger memory sizes (e.g., RMT-64 outperformed RMT-32), Memo achieved ~5% higher success rate on navigation compared to RMT-64. Prior runs with RMT-128 showed unstable convergence, but we plan to include both RMT-128 and RMT-256 in the camera-ready version for completeness.
> > >
> > > A broader observation was that exclusive memory rewriting—without growth—tends to make optimization more difficult. This is because preserving earlier information requires gradients to backpropagate through long chains of memory updates. In contrast, when memory is allowed to grow by appending new summary tokens, earlier representations remain accessible. This enables direct attention to both recent and historical summaries, facilitating more stable and efficient credit assignment. We observed this in both the Mamba results and RMT’s behavior on T-maze, where purely recurrent memory updates led to slower and less stable learning.
> > >
> > >
> > > We believe practical memory systems should grow gradually, with intermittent consolidation, rather than linearly. Balancing flexibility and efficiency is essential, especially in lifelong or streaming settings. Designing mechanisms that enable stable optimization with minimal memory growth remains an open and promising research direction for future work.
> > >
> > >
> > > We sincerely appreciate you raising this discussion, and we’d be happy to address any further questions during the remainder of the discussion period.

---

> > > > ### Comment · Reviewer_SD9F · 2025-08-06
> > > >
> > > > Thank you for conducting the additional experiments, it was interesting to see the results! I maintain my positive view of the paper.

---

### Note · Authors · 2025-08-12

We thank the reviewers for their constructive engagement and positive reception of our work. Across the discussion, all reviewers maintained their positive scores after our clarifications and additional experiments.

Methodological clarity & novelty – Reviewers appreciated that Memo addresses an important and underexplored challenge in RL: memory-efficient long-horizon decision-making. Memo interleaves learnable summary tokens into a Transformer-based RL pipeline, enabling efficient credit assignment and gradient flow across entire histories while substantially reducing inference cost. We clarified differences from Autocompressors (AC) beyond application to RL—Memo operates online in on-policy RL, handles multi-episode cache management, and allows full-context backpropagation where AC uses TBTT.

Baselines & comparisons – In response to SD9F and VTDX, we added new evaluations of Mamba, showing Memo’s superior performance and stability, and explained why recurrent-style memory reuse often slows convergence (T-maze results). We also clarified that our FCT baseline already incorporates StreamingLLM-inspired optimizations, but still degrades in streaming, unlike Memo.

Training efficiency – We addressed concerns on training overhead: Memo is only ~15% slower to train than FCT but achieves 2× faster inference and 10× lower GPU memory at deployment, critical for edge robotics. GPU memory usage during training is comparable to FCT.

We again thank all the reviewers for their positive feedback and constructive suggestions. We will incorporate their recommendations, clarifications, and the new experimental results into the final manuscript.

---

### Decision · Program_Chairs · 2025-09-17

**Decision:**

Accept (spotlight)

**Comment:**

This paper introduces Memo, a transformer based model which incorporates memory tokens for long-horizon, memory intensive tasks. While the paper is an application of ideas from the NLP literature to RL, I think it is still a valuable contribution.

All the reviewers are positive about the paper. All the major concerns, like lack of comparison with Mamba, training efficiency, inference costs, and reasons for instability, are all properly addressed by the authors, and reviewers are satisfied with the answers. It is a good paper that is worth publishing at NeurIPS!